# Selection Signatures in Italian Livestock Guardian and Herding Shepherd Dogs

**DOI:** 10.3390/vetsci10010003

**Published:** 2022-12-21

**Authors:** Arianna Bionda, Matteo Cortellari, Daniele Bigi, Vincenzo Chiofalo, Luigi Liotta, Paola Crepaldi

**Affiliations:** 1Department of Agricultural and Environmental Sciences, Milan University, Via Celoria 2, 20133 Milan, Italy; 2Department of Agricultural and Food Science and Technology (DISTAL), University of Bologna, Viale Fanin 46, 40127 Bologna, Italy; 3Department of Veterinary Sciences, Messina University, Viale Palatucci 13, 98168 Messina, Italy; 4Consortium of Research for Meat Chain and Agrifood (CoRFilCarni), Viale Palatucci 13, 98168 Messina, Italy

**Keywords:** selection signatures, Italian shepherd dogs, SNPs, behavioural genes

## Abstract

**Simple Summary:**

Livestock guardian and herding shepherd dogs are morphologically and behaviourally different, due to the long selection for different tasks made by farmers and breeders. This study aimed to identify genomic regions that best distinguish and characterise four livestock guardian and five herding Italian dog breeds. Genomic SNP data of 158 dogs were compared using two analyses, allowing for the identification of regions harbouring 29 genes. Sixteen runs of homozygosity islands were found in livestock guardians, four of which were partially shared with the fifteen found in herding shepherd dogs. The identified genes were related to dog domestication and behaviour, including herding behaviour, body size and muscle development, the prick or drop ear phenotype, and eye development and functionality. These results contribute to a better understanding of how human selection shaped the genome of dogs selected for different tasks, even considering a limited geographic area.

**Abstract:**

Livestock guardian (LGD) and herding shepherd (HSD) dogs have distinct morphological and behavioural characteristics, long selected by farmers and breeders, to accomplish different tasks. This study aimed to find the genomic regions that best differentiate and characterise Italian LGD and HSD. Genomic data of 158 dogs of four LGD and five HSD breeds, obtained with the 170K canine SNPchip, were collected. The two groups were compared using F_ST_ and XP-EHH analyses, identifying regions containing 29 genes. Moreover, 16 islands of runs of homozygosity were found in LGD, and 15 in HSD; 4 of them were partially shared. Among the genes found that better differentiated HSD and LGD, several were associated with dog domestication and behavioural aspects; particularly, *MSRB3* and *LLPH* were linked to herding behaviour in previous studies. Others, *DYSK*, *MAP2K5*, and *RYR*, were related to body size and muscle development. Prick ears prevailed in sampled HSD, and drop ears in LGD; this explains the identification of *WIF1* and *MSRB3* genes. Unexpectedly, a number of genes were also associated with eye development and functionality. These results shed further light on the differences that human selection introduced in dogs aimed at different duties, even in a limited geographic area such as Italy.

## 1. Introduction

Dogs were domesticated 12,000 to 31,000 years ago, probably as help for hunters to find and capture large prey [1]; however, after the ascent of an agricultural and farming society (11,000–7,000 BC), dogs assumed a new role, becoming fundamental for the management of livestock [2]. Ancient writings suggest that the first shepherd dogs worked primarily as guardians, rather than herders, but it is likely that this distinction was not as clear as today [3]. In fact, at present, shepherd dogs can be distinguished into two main categories: livestock guardian (LGD) and herding shepherd (HSD) dogs.

Livestock guardians are meant to constantly watch and protect the herd from wild predators as well as rustlers. They are usually large, but at the same time fast and strong, in order to deal with large animals such as bears and wolves. To handle the harsh climates that they can live in, they are provided with a thick coat [1,4]. Their attention, trustworthiness, and protectiveness are the result of early socialization with the herd rather than from actual training: around eight weeks of age, they are introduced to the herd, and will live with it full time, sometimes even being breastfed by sheep or goats, thus creating such a strong social bond that they are considered to be part of the flock [1,3,5,6]. Their main characteristic is that they never display predatory behaviour toward the livestock, probably as a consequence of a selection to mature at an early ontogenetic stage, before the emergence of predatory sequences [3,7].

On the other hand, herding shepherd dogs work actively with shepherds, helping them to conduct the livestock and keep animals in a group. To do this, they have to show a hunting motor pattern that is interrupted before the crush-bite-kill sequence [3].

The selection of shepherd dogs has been typically based on their attitude to work: shepherds chose dogs that displayed the best behavioural characteristics, and natural and artificial selection led to the reproductive success of the healthiest and best-adapted dogs [5,7]. Even so, various breeds originated over the decades, and differences also arose among dogs of the same breed, according to the topography of the geographic area they lived and worked in.

Italy counts 17 officially recognized dog breeds, including the Bergamasco shepherd dog and the Maremma and the Abruzzi sheepdog. The first is a herding shepherd dog, with a typical felted coat. Unfortunately, after World War II this breed declined, following a marked decrease in sheep and goat breeding in the Alpine region; thanks to the work of a small group of enthusiasts, the Bergamasco shepherds survived, even though only a small number of dogs are registered every year and they are not diffusely used to work in farms (www.enci.it accessed on 10 December 2022). On the other hand, the Maremma and the Abruzzi sheepdog are livestock guardians most widely used to protect flocks from wild predators. Indeed, several national and international projects provided these dogs to farmers to limit the damage caused by the increasing population of wolves. Furthermore, Italy counts many other local breeds which are in the process of being recognised and should be valorised as well. Among these, we can find several shepherd breeds, such as the Pastore della Sila, the Pastore d’Oropa, the Pastore Apuano, the Pastore della Lessinia e del Lagorai, and the Lupino del Gigante, belonging to Group 1—Sheepdogs and Cattledogs (except Swiss Cattledogs), and the Mannara dog and Fonni’s dog, which are instead placed in Group 2—Pinscher and Schnauzer—Molossoid and Swiss Mountain and Cattledogs. 

In the literature, only a few genomic studies focused on the Italian shepherd dog breeds, mostly based on microsatellites [8,9,10,11,12,13] or on the identification of genes specifically related to sheepdogs [14,15]. Therefore, the aim of the present study was to investigate the genomic regions that best differentiate and characterise Italian herding and livestock guardian shepherd dogs genotyped with a high-density SNP chip.

## 2. Materials and Methods

### 2.1. Animals and Genotyping

The present study included 158 dogs belonging to the following breeds: Maremma and the Abruzzi sheepdog (MARM, n. 20), Mannara dog (MANN, n. 12), Pastore della Sila (SILA, n. 14), and Fonni’s dog (FONN, n. 30), which comprised the livestock guardian group (LGD); Pastore d’Oropa (DORO, n. 15), Pastore Apuano (APUA, n. 19), Bergamasco shepherd dog (BERG, n. 15), Pastore della Lessinia e del Lagorai (PALA, n. 10), and Lupino del Gigante (LUGI, n. 23), which represented the herding shepherd dog group (HSD) (Appendix A).

Blood samples from 26 of the FONN and of all the APUA were collected according to the Ethics Committee’s statement of the University of Messina number 040/2020bis. DNA was extracted according to the recommended manufacturer’s protocol using the DNeasy Blood & Tissue Kit (QIAGEN, Hilden, Germany), and genotyped in outsourcing with a 170K canine SNPchip (CanineHD BeadChip, Illumina, San Diego, CA, USA). Genomic data of all the other dogs came from previous studies [9,16].

### 2.2. Quality Control

Raw genotype data underwent a quality control, performed with PLINK 1.9 software [17]: only those individuals were retained with a call rate ≥ 95% and not directly related to each other (Appendix A), and SNPs with call rates ≥ 95%, with a minor allele frequency (MAF) > 1%, and located on autosomes. The software BEAGLE 4.1 was used for phasing genotype data.

### 2.3. Genomic Analyses

The population structure was assessed through a multidimensional scaling analysis (MDS) of the identity-by-state (IBS) distances using PLINK 1.9 [17]. The genomic backgrounds of all the dogs included in the study were assessed using ADMIXTURE v1.3.0 [18], with a number of clusters (K) ranging from two to eleven: the best-fitting model was identified as the one with the lowest cross-validation value (CV), and individuals’ probabilities of assignment to each K group (Q-value) were analysed.

In order to have homogeneous breed groups and avoid possible bias due to an over-representation of single breeds within a group, before looking for the selection signatures we performed a sample size reduction in the FONN breed in order to have a number of individuals more similar to the other breeds of the LGD group (Appendix A).

The identification of selection signatures in each group of breeds (LGD and HSD) was performed by investigating the runs of homozygosity (ROH), according to the fact that in genomic regions undergoing selection, nucleotide diversity decreases while homozygosity increases around the selected locus [19]. ROH were calculated by applying a sliding window method in PLINK. The sliding window was 50 SNPs long, and no heterozygous SNPs were admitted; ROH was called if a selection (i) consisted of ≥ 50 consecutive homozygous SNPs; (ii) was ≥ 1 Mb long; (iii) had a density ≥ 1 SNP per 50 kb; and (iv) had gaps between two consecutive SNPs that were ≤ 100 kb long. A homozygosity score (H-score) was estimated for each SNP maker as the ratio between the number of its appearances in ROH and the number of the individuals in each group. Only the markers with the top 1% H-scores were considered to identify ROH islands, and were further investigated for the presence of annotated genes in the reference genome CanFam3.1.

Lastly, we investigated the selection signatures that emerged in comparing the two groups, LGD and HSD, using two different and complementary approaches: Wright’s fixation index (F_ST_), using PLINK 1.9 [17], and single SNP cross-population extended haplotype homozygosity (XP-EHH), using SELSCAN 1.1.0 software [20]. While ROH analysis investigated intra-population selection sweep, F_ST_ and XP-EHH are inter-population analyses based on the degree of differentiation between groups due to locus-specific allele frequencies at a single-site (F_ST_) or haplotype (XP-EHH) level. In particular, F_ST_ depends on the proportion of genetic diversity in terms of allele frequency between two populations, thus, detecting the genetic variances that actually underwent divergent selection in the two groups [21], whereas XP-EHH compares the haplotype lengths at each marker between two populations [22], identifying alleles nearly fixated in only one of them. XP-EHH is more powerful in detecting hard sweeps and polygenic selection [23].

In order to minimize the impact of outlier values, the F_ST_ of each SNP was averaged with those of the five adjacent SNPs at both the flanking regions [24]. All of the markers within the top 1% of the empirical distribution both of F_ST_ and normalized XP-EHH values were retained and mapped to CanFam3.1 genome assembly. All of the relevant genes were further investigated in terms of function and related pathways.

## 3. Results

### 3.1. Population Structure

After the procedures of quality control, the dataset consisted of 72 HSD (18 APUA, 11 BERG, 15 DORO, 18 LUGI, and 10 PALA) and 72 LGD (30 FONN, 12 MANN, 16 MARM, and 14 SILA). A total of 120,568 SNPs were retained.

The graphical representation of the first two principal components of the MDS analysis is shown in Figure 1. The two groups are well-separated along the first component (horizontal axis), with LGD on the right and HSD on the left. The second component, instead, isolates DORO from the other dogs. The single breeds are mostly identifiable too, despite partial superimposition of close breeds. 

An admixture analysis was performed, in order to explore the genetic background of all the dogs enrolled in the study. It is interesting to notice that when considering only two structural divisions (K = 2, Figure 2A), all of the LGD showed the prevalence of one cluster (orange, 69 ± 1.4%), while the HSD showed the other cluster (light blue, 79 ± 1.4%); the Q-values for the two clusters were significantly different between the two groups (*p* < 0.0001). The best-fitting model obtained included seven different clusters (K = 7, Figure 2B). A specific cluster (i.e., a prevailing unique colour) was clearly evident for APUA, BERG, DOPO, FONN, LUGI, MARM, and SILA. However, within almost all of the breeds, single dogs showed variable Q-scores for their respective cluster (Appendix A); the introgression of other clusters may reflect historical phylogenetic relationships among the breeds, as well as the effect of unplanned matings that may occur during the traditional transhumance, when these kinds of dogs are not surveilled. Instead, for K = 7, MANN and PALA did not have a unique specific cluster, being the result of a mixture of the other clusters, with MARM prevailing in the MANN and LUGI in the PALA. However, it should be reported that the eighth cluster distinguished MANN. The introgression of MARM could be observed in almost all of the livestock guardians and, to a lesser extent, in PALA. 

### 3.2. Genomic Regions Differentiating Livestock Guardian and Herding Shepherd Dogs

The sample size reduction procedure, applied to make the sizes of the compared breeds uniform, excluded six FONNs. Therefore, the HSD and LGD groups used for selection signature analysis consisted of 72 and 66 dogs, respectively.

The comparison between HSD and LGD led to the identification of 488 SNPs, mapping regions that contained 179 different genes characterised by the top 1% F_ST_ values (0.13−0.31, Figure 3A) and of 393 SNPs, mapping 137 different genes with the top 1% XP-EHH values (2.7–5.1, Figure 3B).

In cross-referencing the results of both analyses, we found 48 SNPs that mapped onto regions containing 29 different genes that were shared between the two (Table 1).

### 3.3. ROH Analysis

The top 1% H-score comprised SNPs located on 16 ROH islands on 11 chromosomes in LGD (H-score: 0.22−0.68), and 15 ROH islands on 12 chromosomes in HSD (H-score: 0.26−0.51); 4 ROH islands, located on chromosomes 1, 13, 25, and 30, were partially shared between the two groups (Table 2). We mapped the SNPs on 43 genes that were harboured in ROH identified in both the groups, 141 in LGD only, and 98 in HSD only (Appendix A). 

## 4. Discussion

Livestock guardian and herding shepherd dogs present several differences in physical appearance and behaviour. For this reason, as it is well known by the shepherds, a dog cannot meet both requirements. In light of this, the main aim of the present study was to determine if these dogs are different in their genome as well and, if so, to identify the regions that diverge the most between them. 

The MDS plot and admixture analyses confirmed that these dogs can also be distinguished from a genetic perspective. This is consistent with Talenti et al. (2018) [9], who investigated the phylogenetic relationship and breed status of several Italian dog populations, finding that Italian livestock guardian and herding shepherd dogs belonged to two different clades; the similarity that we observed between Pastore Apuano, whose SNP data are presented here for the first time, and Lupino del Gigante supports the notion that the first may be allocated in the herding shepherd dog clade as well. It must be said that the geographic localization of these dogs may also have influenced these results, enhancing the differentiation between the Italian herding shepherd dogs, whose cradle is in northern Italy, and the guardian livestock breeds, which instead originated for the most part in central-southern Italy and isles, as already seen in Italian sheep and goats [25,26].

In light of these considerations, our aim was to identify those SNPs and related genes that could better distinguish these two groups, or that may have been specifically selected in one of them. Even though it is for certain that we do not know the amount of variation in specific phenotypes that is explained by most of these markers, it is noteworthy that there is evidence from the literature that some of them are related to behavioural and morphological traits that distinguish HSD and LGD. 

Several of the genes we identified are associated with dog domestication and behavioural or cognitive aspects. For example, in herding shepherd dogs we found ROH on CFA 4 (*RYR2*, *MTR*, and *ACTN2* genes) and 10 (*LLPH* gene), and a large region on CFA 17 that were identified by Kukekova et al. (2018) [27] as differentiating tame and aggressive fox populations developed in the famous Russian farm-fox experiment. Among these genes, four (*ACTN2*, *CCSER1*, *DYSF*, and *NID1*) were also identified as differentiating LGD and HSD in the present study. Moreover, with regards to aggression, mice deficient of *HSF1*, a gene that comprised ROH found in LGD, showed increased offensive aggression towards intruders [28], whereas in dogs, *EXOC4* was associated with the tendency of aggression during a state of nervousness [29], and *MAP2K5* was associated with friendliness toward conspecifics [29]; both genes could distinguish HSD and LGD in our study.

Of particular interest are two genes found in HSD ROH only, namely *MSRB3* and *LLPH*, which have been strongly related to dog herding behaviour in another study [30]. Moreover, some genes that we found on a homozygous region on CFA 22 were highly represented in both LGD and HSD, and harboured *RCBTB1*, *PHF11*, *SETDB2*, *CDADC1*, *CYSLTR2*, and *RCBTB2* genes, which have been already reported by other authors to distinguish hunting and herding dogs [31,32]. 

Guardian and herding purposes can require specific cognitive abilities. For example, herding shepherd dogs strongly rely on communication with the shepherds during their work, while livestock guardians are more independent. In this regard, we found two genes, *NFIA* and *VPS13B*, that have been previously related to communication in dogs [33], and seem to be differentially selected in the two groups or highly homozygous in HSD, respectively. Moreover, the *AGBL4* gene, identified by F_ST_ and XP-EHH analyses, was associated with an ease for dogs to become provoked by uncomfortable or frightening stimuli [29].

Furthermore, other genes differentiating HSD and LGD are related to neurological functionality. Specifically, *DLG2* encodes an excitatory postsynaptic protein involved in the development of striatal connectivity; this gene has been related to autism spectrum disorders in humans, and mice with a deficiency in *DLG2* activity showed decreased sociability and increased stereotypies [34]. Instead, the *RELN* gene encodes reelin, a protein controlling neuronal migration during brain development, and is known to be associated with a number of neurological disturbances in humans; furthermore, a study showed that in mutant zebrafish, the knockout of *RELN* led to a selective reduction in preference for social novelty, and increased serotonin signaling [35]. 

As previously mentioned, despite considerable breed variability, LGD and HSD differ in their appearance: LGD are usually larger, with a protective double coat and floppy ears; on the other hand, most of HSD are medium-sized and display erect, pricked, or folded-over ears [3,36]. The *GABRB1*, *HELB*, and *RELN* genes, which differentiated the two groups in our study, have been previously associated with dog size [29]; moreover, other genes are related with human body height and body mass index: *XPO4*, *SSBP2*, *SH3RF3*, *RGL1*, *RELN*, *PEAK1*, *MAP2K5*, *KIZ*, *HELB*, *DYSF*, *EPB41L1*, *AGBL4*, and *DLG2* (www.genecard.org accessed on 10 December 2022). Moreover, we identified some genes playing a role in muscle development and functionality: *ACTN2* encodes α-actinin-2, which anchors actin filaments at the Z-lines in both skeletal and cardiac muscles [37], and is considered to be a candidate gene associated with physical activity [38]; *DYSF* and *MAP2K5*, instead, encode proteins involved in muscle cell contraction and differentiation. A gene that was frequently contained in ROH of both dog groups was *RYR*, which is related to muscle development in sporting dogs, and was also identified as an adaptation to altitude in the Tibetan Mastiff [39,40,41,42]. Interestingly, *AGBL4*, which differentiated the two groups, was associated to altitude adaptation as well, but in this case, in yak species [43]. Moreover, *CHRM5*, identified in LGD only, is related to the development of muscle mass in dogs [41].

Interestingly, several HSD dogs had ROH on the region of the *WIF1* and *MSRB3* genes, which are related to the pricked/drop-ear phenotype in dogs [29,44,45]. It should be noted that all of the guardian dogs included in this study had drop ears, while all of the herding shepherd dogs had upright ears, with the exception of the semi-drop ears of the Bergamasco shepherd. One of the reasons why the drop ear phenotype tends to be preferred in livestock guardian dogs is that it confers them a more inoffensive appearance [46]. Moreover, it is interesting to note that the so-called “domestication syndrome” is thought to have influenced the rates of ontogenetic processes in dogs, leading to a prolonged duration of the sensitive period of socialization, which is also one of the mechanisms accounting for the suppression of predatory behaviour in LGD, and to the retention of juvenile physical features, including floppy ears [47]. 

Unexpectedly, several of the genes that differentiated the two dog groups are associated with eye development and functionality: *KIZ*, *NFIA*, *AGBL4*, *DLG2*, *DYNC2H1*, and *NID1* (www.genecards.org accessed on 10 December 2022). Particularly, mutations in three of them, *AGBL4*, *DYNC2H1*, and *KIZ*, cause retinal pathologies in humans [48,49,50,51,52], while a *NID1* mutation was found to be the cause of the development of recessive cataracts in Romagnola cattle [53]. These breeds are not commonly screened for eye pathologies. Only one study analysed the epidemiology of inherited ocular disorders in the Maremma sheepdog, finding that over onethird of the dogs were affected by at least one oculopathy; specifically, the most common problems were cataracts, entropion, corneal dystrophy, and retinal dysplasia [54]. It is our opinion that screening for hereditary pathologies should be performed on these breeds; in this way, possible disease predispositions could be identified and controlled, thanks to cautious and targeted selection.

## 5. Conclusions

From a genetic perspective, Italian livestock guardian and herding shepherd dog breeds appear to be well-differentiated. Particularly, some of the most differentiating genes confirm findings of previous studies that compared dogs belonging to different functional groups. Moreover, several genes are associated with behavioural traits, cognitive abilities, communication, and/or neurologic development. Other regions, instead, have been previously associated with body and muscular development, ear shape, or eye functionality. These results shed further light on the differences that human selection introduced in the genome of dogs aimed at different tasks, even within a rather limited geographic area, such as Italy.

## Figures and Tables

**Figure 1 vetsci-10-00003-f001:**
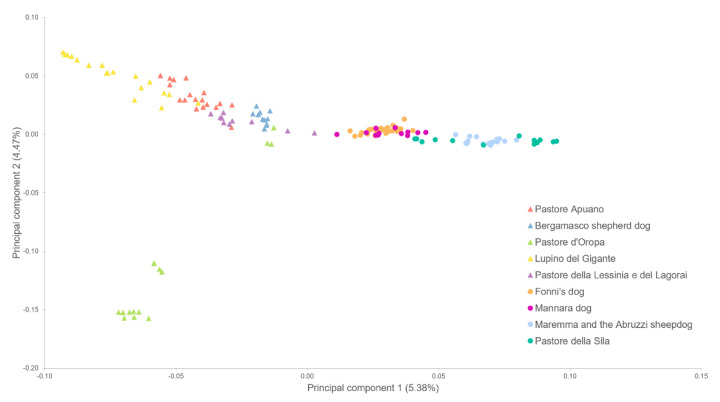
Multidimensional scaling plot representing the first and second principal components. Herding shepherd dogs are represented with diamonds, whereas livestock guardians are represented with circles; each colour indicates a different breed.

**Figure 2 vetsci-10-00003-f002:**
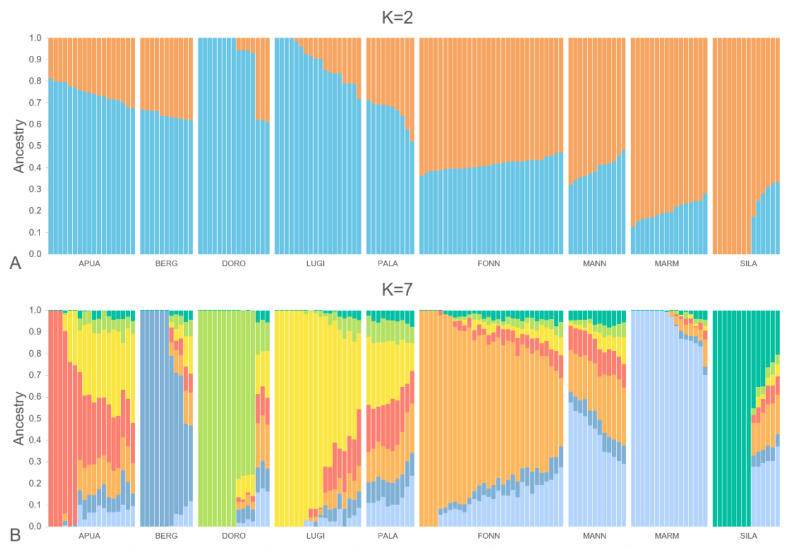
Admixture analysis of all the enrolled individuals, considering a number of clusters (K) equal to two (**A**) and seven (**B**), which resulted in the best fitting model. Each colour represents a different cluster. Herding dog breeds: Pastore Apuano (APUA), Bergamasco shepherd dog (BERG), Pastore d’Oropa (DORO), Lupino del Gigante (LUGI), and Pastore della Lessinia e del Lagorai (PALA); livestock guardian breeds: Fonni’s dog (FONN), Mannara dog (MANN), Maremma and the Abruzzi sheepdog (MARM), and Pastore della Sila (SILA).

**Figure 3 vetsci-10-00003-f003:**
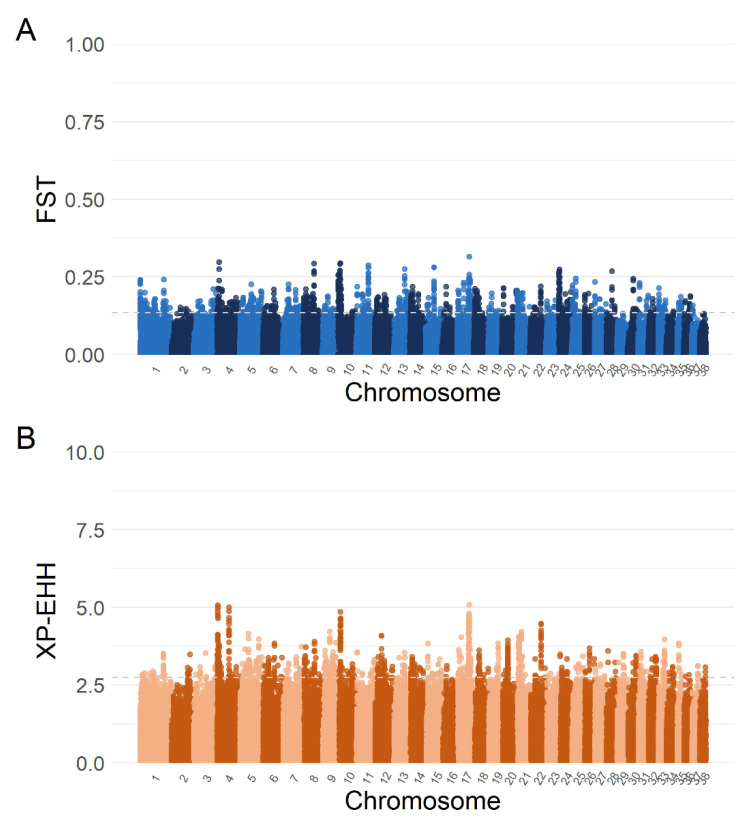
Manhattan plots of Wright’s fixation index (F_ST_) (**A**) and single SNP cross-population extended haplotype homozygosity (XP-EHH) (**B**) analyses comparing livestock guardian and herding shepherd dog breeds. Top 1% F_ST_ absolute values ranged from 0.13 to 0.31, and top 1% log (XP-EHH) from 2.7 to 5.1, as indicated by the grey dashed lines.

**Table 1 vetsci-10-00003-t001:** List of genes shared between Wright’s fixation index (F_ST_) and single SNP cross-population extended haplotype homozygosity (XP-EHH) analyses comparing livestock guardian and herding shepherd dogs.

Gene Symbol	Gene Name	CFA	Start	End
** *SSBP2* **	Single-stranded DNA binding protein 2	3	2,5616,824	25,900,501
** *ACTN2* **	Actinin alpha 2	4	3,348,449	3,415,273
** *NID1* **	Nidogen 1	4	3,935,223	4,017,300
** *TSNAX* **	Translin associated factor X	4	7,937,820	7,967,889
** *CSF1R* **	Colony stimulating factor 1 receptor	4	58,980,788	59,010,510
** *ARHGEF12* **	Rho guanine nucleotide exchange factor 12	5	13,522,896	13,669,155
** *DYNC2H1* **	Dynein cytoplasmic 2 heavy chain 1	5	28,388,664	28,727,334
** *NFIA* **	Nuclear factor I A	5	48,496,773	49,064,848
** *RGL1* **	Ral guanine nucleotide dissociation stimulator like 1	7	17,041,296	17,235,193
** *BAZ1A* **	Bromodomain adjacent to zinc finger domain 1A	8	13,605,182	13,693,847
** *HELB* **	DNA helicase B	10	8,765,617	8,798,092
** *MRPS18A* **	Mitochondrial ribosomal protein S18A	12	12,096,568	12,122,997
** *GABRB1* **	Gamma-aminobutyric acid type A receptor subunit beta1	13	42,992,569	43,349,923
** *CORIN* **	Corin, serine peptidase	13	43,502,665	43,737,813
** *EXOC4* **	Exocyst complex component 4	14	3,334,220	4,077,411
** *SND1* **	Staphylococcal nuclease and tudor domain containing 1	14	8,265,358	8,731,453
** *CSMD2* **	CUB and Sushi multiple domains 2	15	8,090,518	8,110,447
** *AGBL4 ** **	AGBL Carboxypeptidase 4	15	11,914,056	12,427,654
** *DYSF* **	Dysferlin	17	51,011,197	51,228,876
** *RELN* **	reelin	18	16,275,837	16,773,875
** *DLG2* **	Discs large MAGUK scaffold protein 2	21	13,822,304	15,771,954
** *KIZ* **	Kizuna centrosomal protein	24	2,174,905	2,305,811
** *EPB41L1* **	Erythrocyte membrane protein band 4.1 like 1	24	24,925,629	25,0467,76
** *XPO4* **	Exportin 4	25	17,230,031	17,348,031
** *ASUN or INTS13* **	Asunder, spermatogenesis regulator	27	20,479,932	20,507,538
** *MAP2K5* **	Mitogen-activated protein kinase kinase 5	30	31,664,564	31,920,751
** *PEAK1* **	Pseudopodium enriched atypical kinase 1	30	39,134,948	39,207,226
** *CCSER1* **	Coiled-coil serine rich protein 1	32	13,355,503	14,697,519
** *HGD* **	Homogentisate 1,2-dioxygenase	33	24,066,306	24,151,354

SNPs were mapped on CanFam3.1 * In a previous assembly, this SNP was associated with the *BEND5* (BEN domain containing 5) gene.

**Table 2 vetsci-10-00003-t002:** Runs of homozygosity islands found in livestock guardian and herding shepherd dogs.

CFA	Livestock Guardians	Herding Shepherd Dogs
**1**	60,722,335–61,921,241	60,722,335–62,055,218
**4**		2,377,011–3,426,340
**5**	839,609–3,736,188	
**6**		3,107,405–4,085,655
**9**	1,136,406–3,839,981	
**10**	38,721,563–38,882,735 39,511,335–41,893,259	17,278,956–8,658,395
**13**	1,062,829–1,142,946 3,448,621–4,079,768 37,422,915–38,507,976	1,291,574–2,692,593 3,220,205–3,300,233 3,777,508–4,326,026
**14**		3,316,099–4,084,355
**17**	2,578,048–2,795,460	48,593,680–53,104,598
**20**	25,341,988–26,396,630 33,453,310–34,736,398	
**21**		3,247,548–5,676,897 6,926,694–7,041,300
**22**	179,292–5,456,059	707,849–4,454,533
**25**	2,091,732–4,265,859	2,284,963–4,149,337
**27**		9,013,573–10,148,761
**30**	972,855–2,026,353 29,506,940–29,957,952	

## Data Availability

Genomic data can be downloaded from the Gene Expression Omnibus database under accession number GSE121027 (https://www.ncbi.nlm.nih.gov/geo/query/acc.cgi?acc=GSE121027, accessed on 10 December 2022).

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
