# Peer review of "Selection Signatures in Italian Livestock Guardian and Herding Shepherd Dogs"

_vetsci, 2022, doi:10.3390/vetsci10010003_

Round 1
Reviewer 1 Report
English translation checks throughout – things like “homozygosis” would be better as homozygosity. Updates would improve clarity of some statements
It would be helpful to restate that positions in Tables are from CanFam3.1 either in the results or in the table legend(s)
Author Response
English translation checks throughout – things like “homozygosis” would be better as homozygosity. Updates would improve clarity of some statements
AU: Thank you very much for the suggestion, we corrected the translation.
It would be helpful to restate that positions in Tables are from CanFam3.1 either in the results or in the table legend(s)
AU: Thank you very much, we included this information as suggested.

Reviewer 2 Report
Introduction is too long, and not precise enough. I would have liked a more direct intro and not so much historical and details on breeds and breed groups. Only the breeds directly included in THIS STUDY, should have been presented (ref. to lines 92-100
M&M: The data set could have been more directly described in a table. I don't understand why and how the sample size of FONN was reduced (lines 132-134). Reduced from n=30 to 26, however still 30 in line 162. ???
Result section is OK.
Discussion: What is new in this paper. First comes a section, lines 227-237 are explaining that all breeds of the present study have already been investigated, without telling what is new... your results..... line 238-240.
I am generally a bit skeptical to the overemphasize f single SNPs when described as "influencing phenotypic traits without knowing anything about how large part of the VARIATION in these phenotypic traits are explained by the genes or SNPs investigated. I would have moderated the discussion lines 253-274 a bit and underlined that you don't know how large part of the variation is described by this gene or SNP...
.
Author Response
REVIEWER 2
Introduction is too long, and not precise enough. I would have liked a more direct intro and not so much historical and details on breeds and breed groups. Only the breeds directly included in THIS STUDY, should have been presented (ref. to lines 92-100
AU: Thank you very much for your suggestion. We largely reduced the historical part of the introduction (LL 41-51, 62-65, 75-79, and 85-87). However, in our opinion some information about the functional and morphological differences between herding and livestock guardian dogs can be of interest, because it is the main focus of the present research and the rationale behind it. Indeed, we used Italian breeds in order to verify if differences between the two groups are present despite their geographical proximity.
M&M: The data set could have been more directly described in a table. I don't understand why and how the sample size of FONN was reduced (lines 132-134). Reduced from n=30 to 26, however still 30 in line 162. ???
AU: Thank you for your suggestion. We added a supplementary table to better show the datasets used in this paper, specifying the related analyses. Indeed, we firstly performed a quality control on the genomic data in order to keep only SNPs and dogs with high call rate and exclude directly related dogs. This dataset was used in all the analyses concerning the population structure (MDS, Admixture, Genetic distances). When performing the selection signature analyses, instead, we balanced the single breed size, in order to avoid an excessive contribute of the biggest ones (in particular, the Fonni’s dog) in the comparison between livestock guardian and herding dogs: indeed, we aimed to distinguish the overall groups, and keeping an excesive number of Fonni’s dogs, there was the possibility to identify the genes that distinguished herding and Fonni’s dogs only.
In order to clarify this, we specified the datasets used in each analyses in the Table caption and added some details in the Materials and Methods section.
Table S1. Dog breeds included in the present study.
Breed name |
Breed symbol |
Group |
Initial dataset (n.) |
Quality check (n.)* |
Size reduction (n.)** |
Pastore Apuano |
APUA |
Herding dogs (HD) |
19 |
18 |
18 |
Bergamasco shepherd dog |
BERG |
Herding dogs (HD) |
15 |
11 |
11 |
Pastore d’Oropa |
DORO |
Herding dogs (HD) |
15 |
15 |
15 |
Lupino del Gigante |
LUGI |
Herding dogs (HD) |
23 |
18 |
18 |
Pastore della Lessinia e del Lagorai |
PALA |
Herding dogs (HD) |
10 |
10 |
10 |
Fonni’s dog |
FONN |
Livestock guardians (LGD) |
30 |
30 |
24 |
Mannara dog |
MANN |
Livestock guardians (LGD) |
12 |
12 |
12 |
Maremma and the Abruzzi sheepdog |
MARM |
Livestock guardians (LGD) |
20 |
16 |
16 |
Pastore della Sila |
SILA |
Livestock guardians (LGD) |
14 |
14 |
14 |
*This dataset was used for population structure analyses (MDS, genetic distances, and Admixture).
**This dataset was used for selection signature analyses (XP-EHH, FST, and ROH).
Discussion: What is new in this paper. First comes a section, lines 227-237 are explaining that all breeds of the present study have already been investigated, without telling what is new... your results..... line 238-240.
AU: The main aim of the present work was to identify (as far as we know, for the first time) the genomic regions differentiating herding dogs and livestock guardians. Our first step was to understand if genome was actually different between the two groups, and for this reason we performed some population structure analyses. The rationale behind these analyses is different from that of Talenti et al., in which the aim was to determine the breed status of several Italian dog populations by determining the relationship among them and foreign dogs. Moreover, we also included a new breed, the Pastore Apuano, never investigated through SNPdata before.
After establishing that HD and LGD are genomically different, we looked for the regions that diverged the most between them, finding that several of them were related to behavioural and morphological traits that are actually different in the two groups.
In order to clarify this, we rephrased the first part of the discussion as follows:
“Livestock guardian and herding shepherd dogs present several differences in physical appearance and behaviour, so much so that, as it is well known by the shepherds, a dog cannot work both requirements. In light of this, the main aim of the present study was to determine if these dogs are different from a genomic perspective as well and, if so, to identify the regions that diverge the most between them.
The MDS plot and admixture analyses confirm that these dogs can be distinguished also from a genetic perspective. Consistently with Talenti et al. (2018) [11], which investigated the phylogenic relationship and breed status of several Italian dog populations, Italian herding dogs and livestock guardians were classified in two different clades; the similarity that we observed between Pastore Apuano, whose SNP data are presented here for the first time, and Lupino del Gigante supports that the first might be allocated in herding dog clade as well.”
I am generally a bit skeptical to the overemphasize f single SNPs when described as "influencing phenotypic traits without knowing anything about how large part of the VARIATION in these phenotypic traits are explained by the genes or SNPs investigated. I would have moderated the discussion lines 253-274 a bit and underlined that you don't know how large part of the variation is described by this gene or SNP...
AU: we agree that a direct effect of a SNP should not be over-emphasised without validation studies. Following your suggestion, we modulated our considerations in the discussion:
“Even though it is for sure to be considered that we do not know the amount of variation of specific phenotypes that is explained by most of these markers, it is noteworthy that there is evidence from the Literature that some of them are related to behavioural and morphological traits that distinguish HD and LGD.”

Reviewer 3 Report
In the study, the authors screened the selection signatures in Italian livestock guardians and herding dogs with Fst and XP-EHH as well as ROH. And they found that Italian livestock guardian and herding dog breeds could be well differentiated at genomic level and the selection signatures were exhibited by genes associated to behavioral traits, cognitive abilities, communication, and neurologic development. The results of the study are helpful to understand the genetic resource of Italian guardians and herding dogs and facilitate the further study on the functional genes related to the selection signatures.
I have following suggestion for the authors:
1. It will helpful for the readers better understanding the results if the authors could provide the information whether there are overlaps between the genomic regions identified with Fst / XP-EHH and those screened with ROH, and briefly discuss the difference of the results obtained by the different methods.
2. Please change the symbols representing the two types of dogs, and use more distinct symbols, instead of diamond and solid circle, which are hard to be distinguished when they are quite small.
3. For the X-axis of figure 2, please use Chromosome number.
4. Please check the writing of the manuscript. For example, in Line 238, “our aim was to identifies…” should be changed to “our aim was to identify…”
Author Response
REVIEWER 3
In the study, the authors screened the selection signatures in Italian livestock guardians and herding dogs with Fst and XP-EHH as well as ROH. And they found that Italian livestock guardian and herding dog breeds could be well differentiated at genomic level and the selection signatures were exhibited by genes associated to behavioral traits, cognitive abilities, communication, and neurologic development. The results of the study are helpful to understand the genetic resource of Italian guardians and herding dogs and facilitate the further study on the functional genes related to the selection signatures.
I have following suggestion for the authors:
- It will helpful for the readers better understanding the results if the authors could provide the information whether there are overlaps between the genomic regions identified with Fst / XP-EHH and those screened with ROH, and briefly discuss the difference of the results obtained by the different methods.
AU: Thank you for your comment. We added more information about the different and complementary analyses we performed in order to detect selection signatures in the enrolled dog populations: “While ROH analysis investigate intra-population selection sweep, FST and XP-EHH are inter-population analyses based on the degree of differentiation between groups due to locus-specific allele frequency at a single-site (FST) or haplotype (XP-EHH) level. In particular, FST depends on the proportion of genetic diversity in terms of allele frequency between two populations and, thus, detect the genetic variances that actually undergone divergent selection in the two groups [22]; instead, whereas XP-EHH compares the haplotype lengths at each marker between two populations [23], identifying alleles nearly fixated in only one of them, and is more powerful in detecting hard sweeps and polygenic selection [24].
- Please change the symbols representing the two types of dogs, and use more distinct symbols, instead of diamond and solid circle, which are hard to be distinguished when they are quite small.
AU: Thank you very much for the suggestion. We changed the indicator for herding dogs and enlarged the symbols to make them more visible and distinguishable. Hopefully, in the published version of the article, the figure will be even larger than in this draft.
- For the X-axis of figure 2, please use Chromosome number.
AU: Thank you, we modified the figure as suggested.
- Please check the writing of the manuscript. For example, in Line 238, “our aim was to identifies…” should be changed to “our aim was to identify…”
AU: Thank you very much for reporting this to us. We corrected it and checked the whole manuscript.
